# Genome-Wide Identification and Functional Analysis of the GUX Gene Family in *Eucalyptus grandis*

**DOI:** 10.3390/ijms25158199

**Published:** 2024-07-27

**Authors:** Linsi Li, Jiye Tang, Aimin Wu, Chunjie Fan, Huiling Li

**Affiliations:** 1Guangdong Key Laboratory for Innovative Development and Utilization of Forest Plant Germplasm, College of Forestry and Landscape Architectures, South China Agricultural University, Guangzhou 510642, China; 202118110310@stu.scau.edu.cn (L.L.); tangjy.edu@outlook.com (J.T.); wuaimin@scau.edu.cn (A.W.); 2State Key Laboratory of Tree Genetics and Breeding, Key Laboratory of State Forestry and Grassland Administration on Tropical Forestry, Research Institute of Tropical Forestry, Chinese Academy of Forestry, Guangzhou 510520, China

**Keywords:** GUX family genes, *Eucalyptus grandis*, genome-wide identification, functional analysis

## Abstract

Xylan, one of the most important structures and polysaccharides, plays critical roles in plant development, growth, and defense responses to pathogens. Glucuronic acid substitution of xylan (GUX) functions in xylan sidechain decoration, which is involved in a wide range of physiological processes in plants. However, the specifics of *GUXs* in trees remain unclear. In this study, the characterization and evolution of the GUX family genes in *E. grandis*, a fast-growing forest tree belonging to the *Myrtaceae* family, were performed. A total of 23 *EgGUXs* were identified from the *E. grandis* genome, of which all members contained motif 2, 3, 5, and 7. All GUX genes were phylogeneticly clustered into five distinct groups. Among them, *EgGUX01~EgGUX05* genes were clustered into group III and IV, which were more closely related to the *AtGUX1*, *AtGUX2*, and *AtGUX4* members of *Arabidopsis thaliana* known to possess glucuronyltransferase activity, while most other members were clustered into group I. The light-responsive elements, hormone-responsive elements, growth and development-responsive elements, and stress-responsive elements were found in the promoter cis-acting elements, suggesting the expression of GUX might also be regulated by abiotic factors. RNA-Seq data confirmed that *EgGUX02*, *EgGUX03,* and *EgGUX10* are highly expressed in xylem, and *EgGUX09*, *EgGUX10,* and *EgGUX14* were obviously responses to abiotic stresses. The results of this paper will provide a comprehensive determination of the functions of the *EgGUX* family members, which will further contribute to understanding *E. grandis* xylan formation.

## 1. Introduction

Plant evolution has been characterized by the development of complex organs and highly specialized cellular structures, including complex plant cell walls [1]. These secondary cell walls primarily consist of cellulose, hemicellulose, and lignin—three major components that are among the most abundant and renewable resources on Earth [2,3]. These resources can be utilized for the production of biofuels, such as bioethanol and butanol, as well as other high-value biochemicals that are directly or indirectly employed in the fields of food, materials, pharmaceuticals, and chemicals [4]. Xylan is one of the most important hemicelluloses. Based on the secondary cell wall function, xylan biosynthesis-related genes are typically categorized into three groups involved in the synthesis of the main chain, the reducing end, and the side chain (including glucuronide residues, arabinofuranosyl, etc.).

In *A. thaliana*, there are five glucuronic acid substitutions of xylan (GUX, GT8 family)-related genes: *AtGUX1*, *AtGUX2*, *AtGUX3*, *AtGUX4,* and *AtGUX5*. These genes catalyze the attachment of glucuronic acid to the xylan backbone [5,6,7], and the two modifications mediated by *AtGUX1* and *AtGUX2* coexist in the same xylan molecule, which may affect the cross-linking with cellulose or lignin [8]. Previous research conducted by Rennie and colleagues measured glucuronosyltransferase activity in tobacco microcysts by transiently expressing *AtGUX* proteins [9]. Their findings revealed that only *AtGUX1*, *AtGUX2*, and *AtGUX4* exhibited glucuronosyltransferase activity. However, it remains uncertain whether *AtGUX3* and *AtGUX5* are also involved in xylan synthesis. This uncertainty highlights the need for further investigation to fully understand the roles of these genes in plant cell-wall biosynthesis. Golgi proteome analysis in cultured cells of *A. thaliana* has revealed the presence of the *AtGUX3* protein. Since the primary wall of undifferentiated cultured cells constitutes a significant component, it is plausible that *AtGUX3* may also play a role in the synthesis of certain components of the primary wall [10]. Lee et al. conducted an analysis of the expression patterns of the five GUX genes [11]. Their findings indicate that *AtGUX1* and *AtGUX2* are primarily expressed in stems; *AtGUX3* is expressed in roots, stems, leaves, and flowers, and specifically in the xylem of stems; *AtGUX4* is mainly expressed in roots; and *AtGUX5* is primarily expressed in leaves and flowers. Bromley et al. proposed that *AtGUX1* modifies glucuronic acid (GlcA) attachment to xylan with uniformly spaced intervals of 6, 8, 10, or more xylose residues in an even pattern [7]. This modification appears to be the primary mode. In contrast, *AtGUX2* modifies GlcA attachment to xylan with smaller intervals, exercising its modification at intervals of seven or fewer xylose residues. This modification accounts for a smaller proportion and is considered as the secondary mode.

These findings suggest that the GUX family of genes exhibits specialized expression patterns and functional roles in plant cell-wall biosynthesis. *AtGUX3*, despite initial uncertainty regarding its involvement in xylan synthesis, may play a crucial role in the synthesis of the primary wall components in *A. thaliana*. Future studies are needed to further elucidate the precise functions and interactions of these genes in plant cell-wall biosynthesis. The interaction between xylan and cellulose is crucial for determining the mechanical properties of secondary cell walls [12]. Therefore, if the activities of xylosyltransferases, particularly *AtGUX1* and *AtGUX2*, which are pivotal in xylan synthesis, are appropriately downregulated, it could lead to not only normal plant growth and unchanged biomass but also a relative increase in the cellulose content of the plant and a relative decrease in the xylan and lignin content [5,13,14,15]. Modulating the relative xylan decoration process could emerge as an effective strategy for wood modification. To fully harness the efficient utilization of xylan in *E. grandis*, bioengineering techniques can be employed to regulate the structure and proportion of xylan. This approach has the potential to enhance the economic value and sustainability of wood products derived from this species.

*E. grandis*, native to Australia and prevalent in tropical and subtropical regions, is a cornerstone of numerous forestry projects worldwide. China, Brazil, and India, among other countries, boast extensive *E. grandis*-based timber industries that contribute significantly to their economies. This fast-growing tree species offers timber that is easy to process, odor-free, and highly versatile, making it a valuable lignocellulosic raw material. Over the years, *E. grandis* has been extensively studied in various fields, including morphological development and plant resistance. Given its significant economic benefits, researchers have long been interested in understanding and manipulating the biosynthesis and regulation of *E. grandis* timber properties. The ultimate goal is to optimize the growth of *E. grandis* to better meet the diverse economic demands.

In this paper, a comprehensive bioinformatics analysis of the GUX family genes was conducted. This analysis provides a solid foundation for further exploration of the biological functions and regulatory networks of the GUX genes. It lays the groundwork for future research aimed at elucidating the biosynthesis and regulation of timber properties in fast-growing species like *E. grandis*. Additionally, this study opens the door to potential genetic modifications of plant cell walls, offering a promising approach to meeting the diverse demands of the market. By understanding and manipulating the GUX genes, we may be able to optimize the properties of *E. grandis* timber, thus contributing to the sustainable development of the forestry industry. 

## 2. Results

### 2.1. Identification of EgGUXs and Analysis of Physicochemical Properties

Through the BLASTp comparison, 23 GUX proteins were identified, named as *EgGUX01~EgGUX23* (Table 1). Analysis of the physicochemical properties of *EgGUXs* revealed that the amino acid number of each protein ranged from 320 to 645. The *EgGUX04* gene had the highest count (645 amino acids), while the *EgGUX23* gene had the lowest (320 amino acids). Isoelectric point analysis indicated that *EgGUX01* to *EgGUX04* had isoelectrics above 7, while all other members had isoelectric points below 7, suggesting that the *EgGUX* gene family predominantly contained acidic amino acids. The aliphatic amino acid indexes range from 72.16 (*EgGUX23*) to 89.45 (*EgGUX04*), indicating minor differences in thermal stability among the proteins of this family. All *EgGUXs* showed negative values in hydrophilicity analysis, confirming their hydrophilic nature. The most hydrophilic was *EgGUX05*, with a grand average of hydrophilicity (GRAVY) of −0.473, while the least hydrophilic was *EgGUX13*, with a GRAVY of −0.152 (Table 1).

### 2.2. Phylogenetic Tree of GUX Gene Family in E. grandis

To study the evolutionary indications of GUX, the GUX genes in *A. thaliana*, *S. moellendorffii*, *P. patens*, and *P. alba* were named according to the nomenclature and classification of the GUX family genes across different plants by Gallinari et al. [16] (Appendix A). A rooted phylogenetic tree was constructed using TBtools, based on a total of 80 protein sequences from *E. grandis* (23), *A. thaliana* (5), *S. moellendorffii* (14), *P. patens* (18), and *P. alba* (20). Phylogenetic analysis categorized the GUX proteins into five subfamilies, Groups I–V. The *EgGUX* family contained 13, 5, 3, and 2 members from Group I to Group IV, respectively, with no *EgGUXs* members in Group V. Given the demonstrated glucuronosyltransferase activity in *AtGUX1*, *AtGUX2*, and *AtGUX4*, it is inferred that *EgGUX* family members are closely related to *A. thaliana AtGUX1*, *AtGUX2*, and *AtGUX4* corresponding to *EgGUX01*~*EgGUX05* in Group III and Group IV. Furthermore, the phylogenetic tree suggests that *EgGUX* genes have closer affinities with *P. alba* and *A. thaliana*, and more distant affinities with *P. patens* and *S. moellendorffii* (Figure 1).

### 2.3. Analysis of EgGUX Structures and Conserved Domains

According to protein-conserved motifs, the phylogenetic tree of the 23 *EgGUXs* was clustered into three classes (Figure 2A). The conserved motifs of the 23 *EgGUXs* were then presented using the MEME online program (Figure 2B). It is evident that the amino terminus of the 23 GUX proteins generally contains relatively conserved structural domains. Both classes I and III consist of motifs 1 to 7 with identical ordering, while motifs 8 to 10 are found in class II; the members of the *EgGUX* family in class II also exhibit identical ordering. The structural organization of the genes, based on coding sequences (CDSs) and untranslated regions (UTRs), was analyzed using TBtools (Figure 2C). The results show that the distribution of introns varies among individual genes within the *EgGUX* members.

### 2.4. Chromosomal Localization of GUX Gene Family Members

A large proportion (34%) of genes in the *E. grandis* genome were expanded by tandem duplications [17]. Twenty genes were located on four chromosomes according to the annotation file, and three genes were not presented on the chromosomes but on three different scaffolds (Figure 3). These 20 genes are distributed in various patterns, including both clustered and independent distribution. Notably, 11 *EgGUXs* formed a high-density gene island clustered on Chr02.

### 2.5. Analysis of Intraspecific and Interspecific Collinearity in the EgGUX Family Genes

Events through genome-wide, segmented, dispersed, or tandem gene duplication are considered major drivers of evolution [18]. To analyze the genome-wide duplication of the *EgGUXs*, members of the *E. grandis* GUX gene family were subjected to collinearity analysis (Figure 4a). The results revealed that this gene family undergoes self-replication at a very low frequency in *E. grandis*, with only one pair of tandem and segmental duplications, *EgGUX13* and *EgGUX14*.

Further analysis of interspecific co-linkage relationships using the *A. thaliana* and *P. alba* genomes showed (Figure 4b) that 8 co-linkage pairs were formed between the *A. thaliana* and *E. grandis* genomes, and 17 co-linkage pairs were formed between the *P. alba* and *E. grandis* genomes. This suggests that *P. alba* and *E. grandis* may have closer phylogenetic relationships. 

### 2.6. Analysis of Cis-Acting Elements of the Promoter on the EgGUX Gene Family

Gene expression can be regulated by the binding of transcription factors (TFs) to cis-elements within their promoter regions [19]. The cis-acting elements in the 2000 bp promoter region upstream of the *EgGUXs* were analyzed together with phylogenetic tree (Figure 5). Out of the 22 *EgGUXs*, 22 response elements were selected and predicted, except for *EgGUX21*, which lacks promoter regions (Appendix A). These were divided into four categories: light-responsive elements, hormone-responsive elements, growth and development-responsive elements, and stress-responsive elements. Light response elements were the most numerous, and almost *EgGUXs* include hormone-responsive elements, including those responsive to auxin, abscisic acids, gibberellin, MeJA, and SA. Additionally, there were seven types of growth and development response elements and six types of stress response elements. 

### 2.7. Expression of EgGUXs in Different Tissues of E. grandis

To gain a deeper understanding of the expression patterns of *EgGUXs* in different tissues of *E. grandis*, expression profiles were conducted. These tissues included 6−month−old young leaves, adult leaves, xylem, phloem, roots, and the stem apex. The results revealed varying degrees of expression across all tissues. In adult leaves, the expression of *EgGUX09*, *EgGUX10*, *EgGUX13*, and *EgGUX14* was higher (Figure 6a). In contrast, in stems, the expression of *EgGUX02*, *EgGUX03*, *EgGUX04*, *EgGUX05*, *EgGUX10*, and *EgGUX13* was significantly higher in the third, fifth, seventh, ninth, and eleventh internodes (Figure 6b). In 6−year−old trees, only *EgGUX02*, *EgGUX03,* and *EgGUX10* showed higher expression in xylem, while the expression of other *EgGUXs* in other tissues was relatively low (Figure 6c).

### 2.8. Expression of EgGUX Genes under Abiotic Stress and Phytohormone Treatments

To explore the differential expression of *EgGUX* genes in response to abiotic stress and phytohormone treatments, shoots of 2−month−old *E. grandis* were exposed to boric acid deficiency, phosphorus deficiency, and salt stress, while the leaves were sprayed with SA and MeJA. The raw expression data represented by the circular graphs revealed that *EgGUX09*, *EgGUX10*, and *EgGUX14* exhibited higher expression levels under conditions of boric acid deficiency, phosphorus deficiency, salt stress, SA treatment, and MeJA treatment. Row normalization using the rectangle graphs technique revealed that the expression of *EgGUX09*, *EgGUX10*, and *EgGUX14* showed significant differences with increased treatment. Additionally, the differential expression of *EgGUX02*, *EgGUX03*, *EgGUX04*, and *EgGUX05* was also more pronounced (Figure 7).

### 2.9. 3D Structure Analysis of E. grandis GUX Gene Family Members

After utilizing SWISS-MODEL for homology modeling and SAVES for evaluation, the 3D structures of several representative *EgGUX* proteins were successfully obtained (Figure 8). The 3D structures within the same subfamily exhibit considerable similarity, with minimal differences (Figure 8). However, when comparing structures from different subfamilies, more pronounced differences are observed. These structural variations primarily arise from the diverse lengths of α-helices, β-turns, and irregular convolutions present in each protein. These differences in structural elements lead to variations in the spatial folding angles, which in turn may underlie the distinct functional roles performed by these proteins.

## 3. Discussion

With the development of molecular biology techniques, the GT8 gene family has been identified in detail in an increasing number of plants, such as cotton and sugarcane. These studies have played an important role in cell-wall hemicellulose modification. The GT8 family has been associated with the biosynthesis and modification of plant cell walls [20], and there exists a functional divergence. Specifically, the glucuronosyltransferase (GUX) family catalyzes the binding of glucuronide (GlcA or MeGlcA) to xylan side chains [5,7]. In model plants, the GUX genes are involved in decorating cell wall hemicellulose by adding glucuronic acid substitutions to the xylan skeleton. Additionally, the irregular xylem (IRX) genes IRX8 and galacturonosyltransferase-like 1 (PARVUS) also belong to the GT8 family, which functions on the tetresacharride reducing end of xylan [6,8,11].

Mutations in the GUX genes increase the enzyme’s affinity for cell wall polysaccharides, thereby reducing biomass recalcitrance in *A. thaliana*, which is of economic importance to the biotechnology industry. Mutations in GUX genes were reported to reduce the presence of such residues, hindering the access of cellulases to biomass and increasing saccharification yield [5,11,21].

Since glucuronic acid is negatively charged, an increase in glucuronic acid can help hemicellulose to better separate from cellulose, resulting in producing more cellulose nanofibrils (CNFs) [22]. CNFs have great potential in papermaking, packaging, optoelectronics, and other fields because of their good biocompatibility, reproducibility, high mechanical properties, good hydrophilicity, and high light transmittance. By analyzing the role of the GUX gene family in Eucalyptus, using CRISPR-Cas9, overexpression, and other technologies for molecular breeding, and regulating the synthesis efficiency of the glucuronic acid side chain and improving the chemical components of lignocellulosic raw materials, we can utilize CNFs to realize efficient industrial application and provide a new idea for genetic improvement in Eucalyptus.

In this study, we determined that the number of GUX proteins in *E. grandis* (23) is approximately five times the number of GUX proteins in *A. thaliana* (5) [9]. This suggests that the GUX gene family has significantly expanded throughout evolution in *E. grandis*, which is consistent with the number of 31 GUX proteins in cottons [23]. Furthermore, our analysis of the physicochemical properties of these 23 proteins revealed that more than half of them exhibited structural instability. Based on the numbers of these gene families, we speculated that the GUX genes might be increase in more lignocellulose plants.

The phylogenetic tree analysis of *E. grandis*, *A. thaliana*, *S. moellendorffii*, *P. patens*, and *P. alba* indicates that the *EgGUX* proteins of *EgGUX01~EgGUX05* are closely related to *AtGUX1*, *AtGUX2*, and *AtGUX4*, which have been definitively shown to possess glucuronosyltransferase activity. All of these belong to Group III. It is presumed that these three *EgGUX* proteins have a higher likelihood of glucuronosyltransferase activity. These results indicated that these *EgGUX* proteins might function on add GlcA into xylan side-chain. Meanwhile, the expression heatmap derived from RNA-Seq data revealed that the *EgGUX10* and *EgGUX14* were most prominently expressed in 6−month−old *E. grandis* young leaves, adult leaves, xylem, phloem, and stem apexes. *EgGUX10*, *EgGUX02*, *EgGUX03*, *EgGUX13*, *EgGUX04*, and *EgGUX05* were highly expressed in the 3rd, 5th, 7th, 9th, and 11th stem nodes of 6−month−old grown *E. grandis*, as well as in the xylem and phloem of 6−year−old *E. grandis*. This could be due to the greater demand for xylans and more complicated synthesis mechanisms in woody plants, so other GUX genes can also express highly. 

Cis-elements play a crucial role in the regulatory network controlling plant growth and development. The promoter regions of almost all *EgGUX* genes contain cis-elements that may be responsive to hormones such as auxin, ABA, gibberellin, SA, and MeJA. These cis-elements are essential for plant growth and development and stress response, suggesting that the *EgGUX* gene family may be involved in growth, development, and resistance in *E. grandis*. 

Based on RNA-Seq data, the heatmap analysis revealed that several members of the *EgGUX* family were significantly expressed in the phloem and xylem, while others were prominently expressed in leaves. Overall, *EgGUX* members demonstrated stronger expression in stems, xylem, and adult leaves. This could be attributed to the higher presence of secondary cell walls in stems of Eucalyptus, requiring increased glucuronide xylan for growth and development. When exposed to abiotic stress and phytohormone treatments, *EgGUX09*, *EgGUX10*, and *EgGUX14* exhibited higher expression levels and more significant differential expression patterns. Meanwhile, the expressions of *EgGUX01~EgGUX05*, which are more closely related to the *AtGUXs*, did not change regularly with treatment time. This suggests that the *GUXs* have undergone evolutionary and functional divergences.

## 4. Material and Methods

### 4.1. Plant Materials

In this study, the plant material utilized was the *E. grandis* clone GL1. The clone was cultivated from a single plant that showed cold-resistance characteristics during a cooling disaster in Jiangxi Province, China, in 2005. The researchers propagated the tissue culture seedlings of the single plant in April 2006, and named the clone GL1 [24]. The rooted shoots of the in vitro plants were cultivated in a controlled environment at the Research Institution of Tropical Forestry (located at E 113.385°, N 23.191°), which is part of the Chinese Academy of Forestry in Guangzhou, China. Tissue samples, including root, mature and young leaves, xylem, and phloem, were harvested from 6−month−old shoots grown in the greenhouse of the same institution. Additionally, various stem internodes—specifically the 1st (shoot apex), 3rd, 5th, 7th, 9th, and 11th—were collected from these 6−month−old shoots. Organs and tissues such as xylem, phloem, and cambium were also gathered from 6−year−old trees grown at the Zhenshan nursery (located at E 112.673°, N 23.330°) in Sihui, Zhaoqing, Guangdong, China.

For the nutrient deficiency treatment, 2−month−old plants were subjected to boron deficiency and phosphorus deficiency conditions using half-strength Hoagland’s solution. The pH of the solution was adjusted to 5.8 using NaOH. Following the exposure period, the roots were carefully blotted, immediately frozen in liquid nitrogen, and stored at −80 °C. 

For hormonal and stress challenge experiments, shoots were propagated through tissue culture in pots within the greenhouse for a duration of 2 months. Fully expanded young leaves located beneath the apex, ranging from four to eight in number, were selected for hormone and stress treatments. During this period, the shoots attained a length of 25−35 cm. Hormone treatments involved spraying the leaves with 100 μM salicylic acid (SA) and methyl jasmonate (MeJA). Subsequently, the treated leaves were sampled at 0, 1, 6, 24, and 168 h post-treatment. A salinity stress treatment was administered by irrigating the plants with 200 mM NaCl, followed by leaf sampling at the same time intervals. To ensure representation, at least three individual plants were pooled to represent one biological replicate, and triple repeats were performed for different organs and treatments.

### 4.2. Identification of EgGUX Gene Family Members and Analysis of Physicochemical Properties

The *AtGUX* gene sequence was initially obtained by downloading it from the TAIR database (http://www.arabidopsis.org, accessed on 7 January 2024). The *A. thaliana* GUX gene sequence was used as a probe, with the threshold set at e^−10^ [25], and the homologous sequences were retrieved by BLASTp searching in the *E. grandis* genome database, thus obtaining the *E. grandis* GUX gene sequence. Physicochemical properties such as amino acid count, isoelectric point, and molecular weight of the *E. grandis* GUX family members were analyzed using ProtParam tool provided by the ExPASy website (https://web.expasy.org/cgi-bin/protparam/protparam, accessed on 16 January 2024) [26]. ProtScale tools also can be used to analyze the hydrophilic protein found in ExPASy website (https://web.expasy.org/protscale/, accessed on 16 January 2024). 

### 4.3. Phylogenetic Tree Construction of the GUX Family Genes in E. grandis

In order to study the phylogeny of the GUX gene family, four species, *A. thaliana*, *Selaginella moellendorffii*, *Physcomitrium patens*, and *Populus alba* were selected for sequence comparisons and sequence similarity analyses with the *EgGUXs*. Multiple sequence comparisons were performed, and sequence similarities were determined, with the dicotyledonous *A. thaliana* genome annotated from The *Arabidopsis thaliana* Information Resource. The genome annotation information for *S. moellendorffii* and *P. patens* was obtained from NCBI (https://www.ncbi.nlm.nih.gov/, accessed on 18 January 2024), and the genome annotation information for *P. alba* was obtained from ‘Science China Life Sciences’ [27]. Homologous proteins from these five species were screened using TBtools software (v2.110) to construct a phylogenetic tree using ML (maximum-likelihood method) with parameters set to UltraFast BootStrap, Bootstrap Number:5000 (Model:Auto, Number of Treads:Auto;). Other parameters were set to default values [28], and the resulting evolutionary tree was classified and annotated using the iTOL (https://itol.embl.de/, accessed on 25 January 2024) website [29].

### 4.4. Analysis of E. grandis GUX Gene Structure and Conserved Structural Domains

Multiple protein sequence comparisons of *EgGUXs* were performed using Clustal W in MEGA11 software (v11.0.13) [30], with the parameters set to default values. The results of these comparisons were used to construct a phylogenetic tree using the maximum-likelihood method (ML), with parameters set to UltraFast BootStrap, Bootstrap Number: 5000, Model:Auto, Number of Treads:Auto; other parameters were set to default values. All *E. grandis* GUX protein family members were classified according to the results of the phylogenetic tree. Conserved motifs of GUX proteins were analyzed using the MEME website (http://meme-suite.org/index.html, accessed on 5 January 2024) [31], with a *p*-value of <1e^−5^ for each motif. Gene structures were analyzed using the TBtools software (v2.110), and both the phylogenetic tree and conserved motifs of the GUX proteins were visualized.

### 4.5. Chromosomal Localization of E. grandis GUX Gene Family Members

The gff3 file in the *E. grandis* genome was used in Tbtools software to obtain the staining and positioning information of the *EgGUX* family members and draw the distribution map of the genes on the chromosome. TBtools was used to analyze the distribution of the *EgGUX* genes on chromosomes and the gene density of each chromosome and to create a map. In the ‘One Density Profile’ feature, the parameter “Bin Size” was selected as 1,000,000, with default values used for the remaining parameters. 

### 4.6. Intraspecific and Interspecific Collinearity Analysis of the E. grandis GUX Family Genes

All genes in the *E. grandis* genome were compared with each other using TBtools software with default parameters. After importing GC base content, gene density, and the number of undetermined nucleotides, the comparison results were visualized by Advanced Circos. To further analyze the evolutionary relationships of the GUX family genes, collinearity analysis was conducted using the GUX family genes from *P. alba* and *A. thaliana*. The results were analyzed and plotted using the Dual Systeny Plot feature in TBtools software. 

### 4.7. Analysis of Cis-Acting Elements in the Promoter of the EgGUXs Family

The 2 kb upstream sequence of *EgGUXs* was extracted using the ‘GXF Sequence Extraction’ tool in TBtools software [32]. It was then analyzed and compared with the online PlantCare database (https://bioinformatics.psb.ugent.be/webtools/plantcare/html/, accessed on 20 January 2024) to identify and retrieve cis-acting elements of the gene family. These elements were then combined with the phylogenetic tree of *E. grandis* and visualized using the ‘Basic BIOsequence View’ tool in TBtools.

### 4.8. The Expression Patterns of GUX Gene Family in E. grandis

To better understand the role of GUX genes in *E. grandis*, gene expression patterns were analyzed using our previous gene expression data [33]. The plant materials included roots, adult and young leaves, xylem, and phloem from 6−month−old plants, as well as the 1st (shoot apex), 3rd, 5th, 7th, 9th, and 11th internodes from 6−month−old branches. Additionally, xylem and phloem were sampled from 6−year−old *E. grandis*. The expression of GUX genes was examined in these different tissues to explore their expression patterns across various tissues and growth stages. Using the TBtools Heatmap function, transcriptome data from the 6−month−old branches were log-normalized using the formula ‘logbase (value + LogWith)’ (where base is 2.0 and LogWith is 1.0) to minimize data dispersion. Subsequently, ‘Cluster Rows’ was selected to perform hierarchical clustering. A heatmap depicting the expression patterns was generated following the hierarchical clustering.

Differential expression analysis was then conducted on young leaves treated with SA, MeJA, and salt stress for 0, 1, 6, 24, and 168 h. Employing the TBtools Heatmap function, the transcriptome data were log-normalized by selecting ‘Log Scale’. The ‘Tile Shape’ was set to ‘Circle’ and ‘Scale Size By Area’ was selected to create a raw expression heatmap. Additionally, after log-normalization, ‘Row Scale’ was chosen to normalize the rows, resulting in a heatmap that more clearly highlights differential expression. The two heatmaps were then combined to present a comprehensive picture of GUX gene family expression in response to abiotic stress and phytohormone treatment.

### 4.9. Protein Structure Prediction of EgGUXs

The 3D structures of *EgGUX* family proteins were constructed by homology modelling using the SWISS-MODEL (https://swissmodel.expasy.org/, accessed on 27 January 2024) online tool [34].

## Figures and Tables

**Figure 1 ijms-25-08199-f001:**
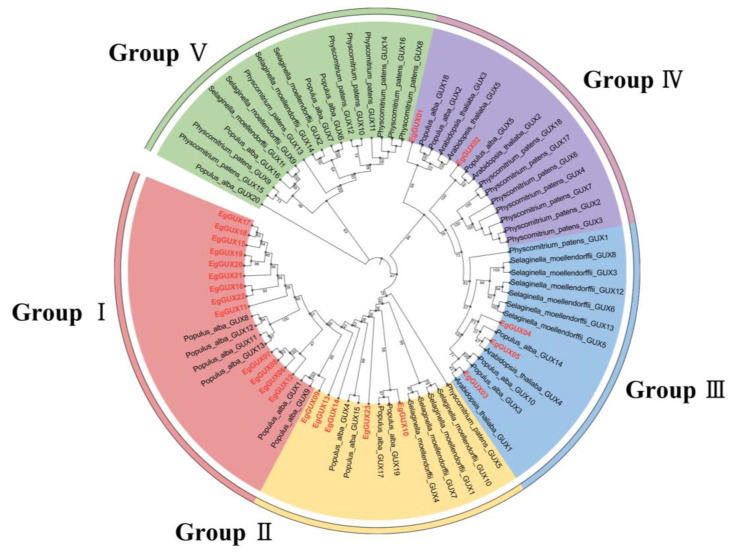
Phylogenetic analysis of GUX genes. the *GUXs* in *E. grandis* (23), *A. thaliana* (5), *S. moellendorffii* (14), *P. patens* (18), and *P. alba* (20) were picked up. A maximum-likelihood phylogenetic tree was generated by MEGA11 with full-length GUX sequences (5000 bootstrap replicates). The five groups are highlighted with red, yellow, green, blue, and purple, respectively.

**Figure 2 ijms-25-08199-f002:**
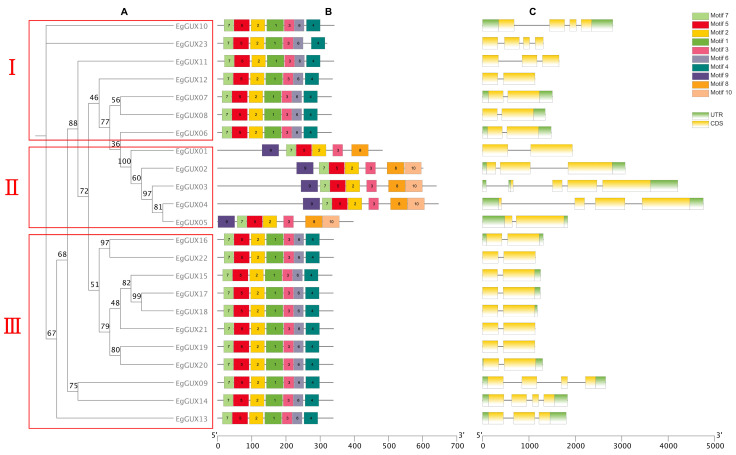
Predicted *EgGUX* protein phylogeny, conserved amino acid motifs, and gene structure. (**A**) Rooted maximum-likelihood phylogeny of *EgGUX* proteins, showing subfamily classification. (**B**) Motif compositions of GUX in *E. grandis* are presented in different colors ranging from motif 1 to 10. (**C**) Position of exons and introns in the *EgGUX* gene models. At the bottom of the figure, the relative position is proportionally displayed based on the kilobase scale.

**Figure 3 ijms-25-08199-f003:**
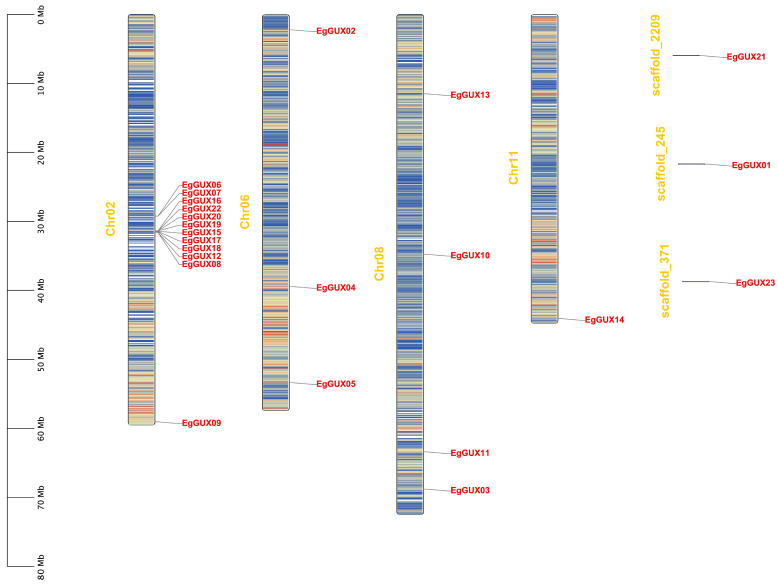
Chromosomal locations of *EgGUX* genes. The chromosome number is shown on the middle of each chromosome. The scale bars represent the length in mega-bases (Mb). Red areas of each chromosome indicate high gene density. Blue areas indicate low gene density.

**Figure 4 ijms-25-08199-f004:**
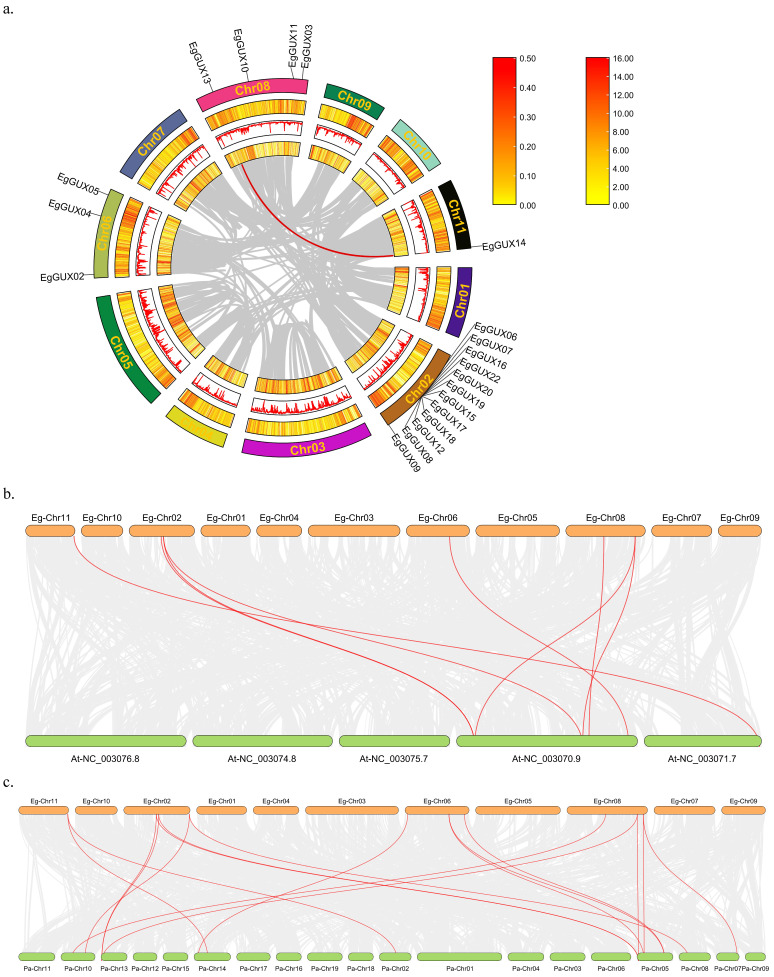
Collinearity relationship of GUX family members of *E. grandis*, *A. thaliana*, and *P. alba*. Intraspecific collinearity relationship of GUX family members of *E. grandis* (**a**) The gray line indicates all collinearity blocks in the *E. grandis* genome, and the red line indicates the segmental repeats of the *EgGUX* genes. Interspecies collinearity relationship among *E. grandis* and *A. thaliana* (**b**) and *E. grandis* and *P. alba*. (**c**) The gray line indicates all collinearity blocks in the genomes, and the red line indicates the collinearity of GUX genes.

**Figure 5 ijms-25-08199-f005:**
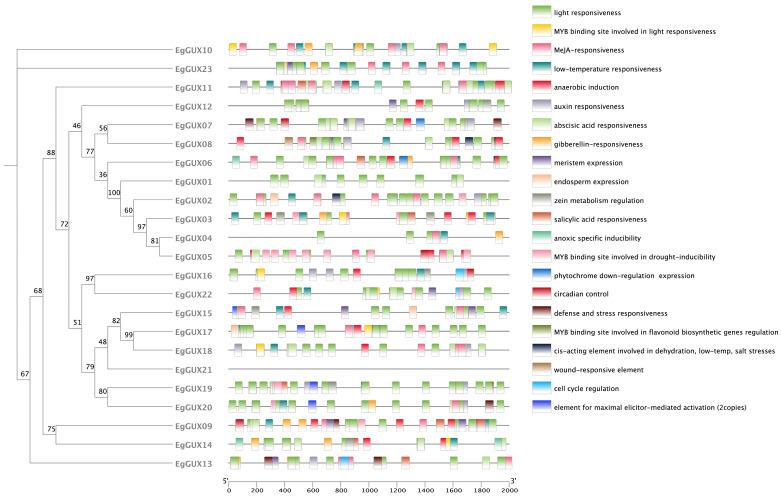
Analysis of cis-acting elements in promoters of GUX gene family in *E. grandis*. The 22 squares on the right represent the various cis-acting elements of the promoter.

**Figure 6 ijms-25-08199-f006:**
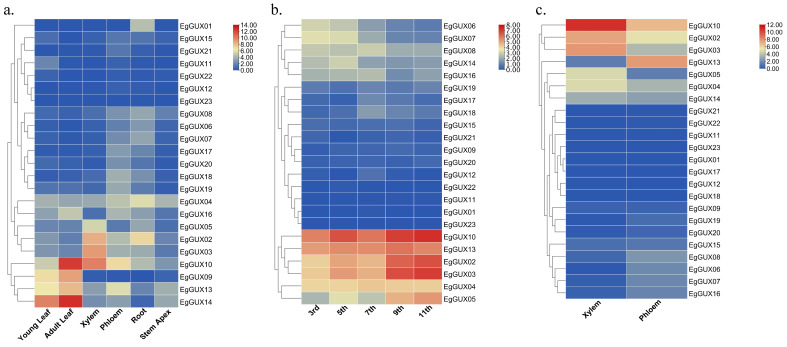
Heatmap of *EgGUX* gene expression in different tissues. (**a**) Heatmap of expression in 6−month−old young leaves, adult leaves, phloem, xylem, roots, and stem apex. (**b**) Expression heatmap in 6−mont−old tree of 3rd, 5th, 7th, 9th, and 11th internodes. (**c**) Expression heatmap in 6−year-old xylem and phloem of *E. grandis*. Expression values based on RNA-seq are shown from blue to red, indicating low to high expression.

**Figure 7 ijms-25-08199-f007:**
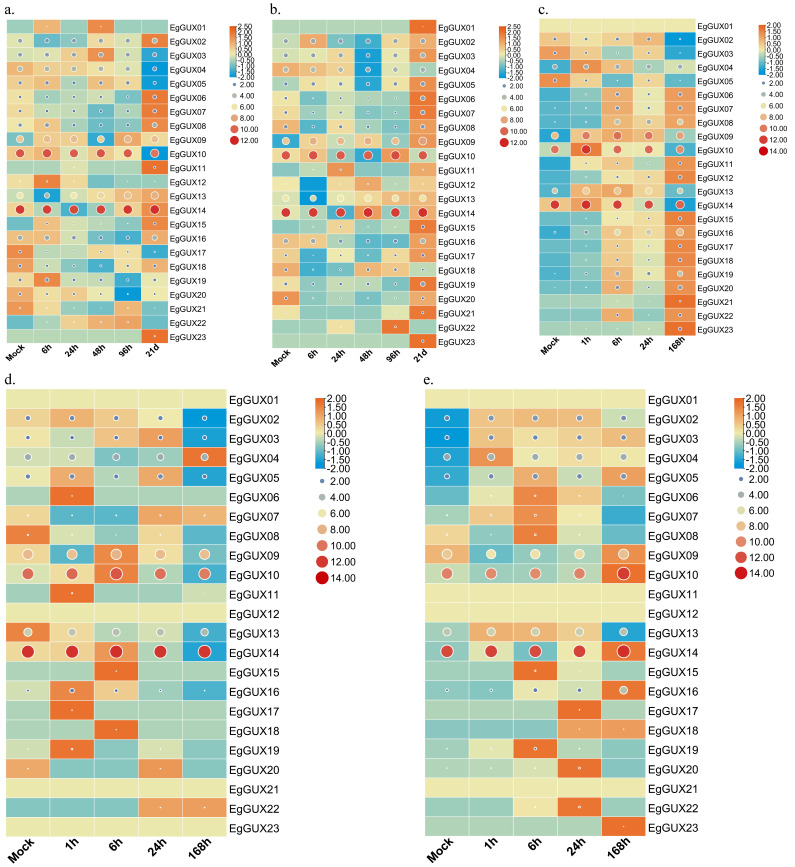
Heatmap of *EgGUX* gene expression under abiotic stress and phytohormone treatment. (**a**) Heatmap of expression in 2−month−old shoots at 0, 6, 24, 48, 96 h, 21 d of boron deficiency treatment. (**b**) Heatmap of expression in 2−month−old shoots at 0, 6, 24, 48, 96 h and 21 d of phosphorus deficiency treatment. (**c**) Heatmap of expression in 2−month−old young leaves under salt stress treatment for 0, 1, 6, 24, 168 h. (**d**) Heatmap of expression in 2−month−old young leaves under SA treatment for 0, 1, 6, 24, 168 h. (**e**) Heatmap of expression in 2−month−old young leaves under MeJA treatment for 0, 1, 6, 24, 168 h. Scaled log2 expression values based on RNA-Seq data are shown from blue to red, indicating low to high expression. The size and color of the circles represent the original numerical size of the *EgGUXs* protein expression. The larger the circle and the redder the color, the higher the expression; the smaller the circle and the bluer the color, the lower the expression. The rectangle graphs were row normalized, with bluer colors indicating lower expression at that treatment length, and more orange colors indicating higher expression at that treatment length.

**Figure 8 ijms-25-08199-f008:**
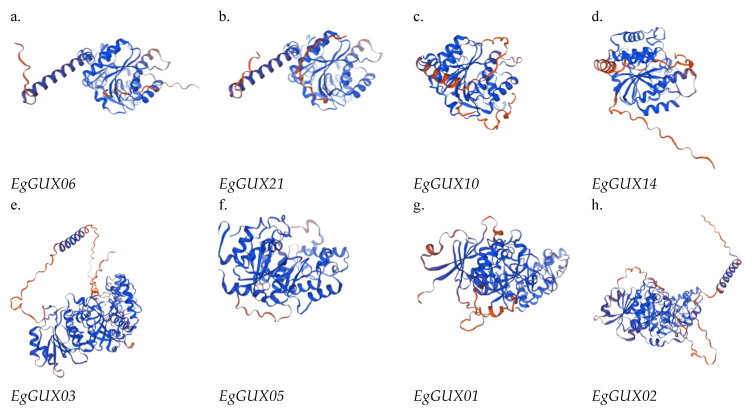
3D structure analysis of *EgGUX* gene family members. (**a**) Three-dimensional structure of subfamily I *EgGUX06* proteins. (**b**) Three-dimensional structure of subfamily I *EgGUX08* proteins. (**c**) Three-dimensional structure of subfamily II *EgGUX10* proteins. (**d**) Three-dimensional structure of subfamily II *EgGUX14* proteins. (**e**) Three-dimensional structure of subfamily III *EgGUX03* proteins. (**f**) Three-dimensional structure of subfamily III *EgGUX05* proteins. (**g**) Three-dimensional structure of subfamily IV *EgGUX01* proteins. (**h**) Three-dimensional structure of subfamily IV *EgGUX02* proteins.

**Table 1 ijms-25-08199-t001:** Physicochemical of proteins encoded by the GUX gene family in *E. grandis*.

Gene Name	Gene ID	Number of Amino Acids	Molecular Weight	Theoretical pI	Instability Index	Aliphatic Index	Grand Average of Hydropathicity
*EgGUX01*	Eucgr.L01540.1.v2.0	482	55,737.14	9.49	40.22	78.86	−0.428
*EgGUX02*	Eucgr.F00232.1.v2.0	600	69,750.06	8.92	60.20	86.92	−0.391
*EgGUX03*	Eucgr.H04942.1.v2.0	639	73,737.56	8.07	42.14	89.45	−0.341
*EgGUX04*	Eucgr.F02737.1.v2.0	645	74,908.79	7.60	41.35	85.30	−0.398
*EgGUX05*	Eucgr.F04263.1.v2.0	396	46,149.31	6.02	36.47	79.39	−0.473
*EgGUX06*	Eucgr.B01793.1.v2.0	332	38,282.92	5.64	30.85	83.10	−0.324
*EgGUX07*	Eucgr.B01791.1.v2.0	332	38,367.07	6.12	30.34	84.28	−0.323
*EgGUX08*	Eucgr.L00234.1.v2.0	332	38,445.19	5.64	29.03	83.98	−0.296
*EgGUX09*	Eucgr.B03987.1.v2.0	337	38,222.72	4.93	43.28	79.85	−0.200
*EgGUX10*	Eucgr.H02584.1.v2.0	340	38,511.16	5.06	43.07	81.44	−0.161
*EgGUX11*	Eucgr.H03312.1.v2.0	339	38,613.92	5.34	46.82	74.54	−0.397
*EgGUX12*	Eucgr.L00235.1.v2.0	335	38,280.61	4.93	43.02	79.43	−0.348
*EgGUX13*	Eucgr.H00906.1.v2.0	337	38,339.02	5.07	42.72	82.79	−0.152
*EgGUX14*	Eucgr.K03563.1.v2.0	337	38,407.05	4.89	41.62	86.47	−0.177
*EgGUX15*	Eucgr.L00245.1.v2.0	334	38,516.01	5.07	30.26	80.27	−0.276
*EgGUX16*	Eucgr.L00251.1.v2.0	338	38,950.77	5.45	31.96	82.81	−0.288
*EgGUX17*	Eucgr.L00243.1.v2.0	337	38,709.39	5.23	32.79	81.07	−0.267
*EgGUX18*	Eucgr.L00240.1.v2.0	337	38,775.52	5.23	31.22	83.92	−0.230
*EgGUX19*	Eucgr.L00248.1.v2.0	337	38,594.22	5.24	27.89	80.74	−0.266
*EgGUX20*	Eucgr.L00249.1.v2.0	337	38,664.44	5.45	27.40	81.90	−0.261
*EgGUX21*	Eucgr.L03185.1.v2.0	338	38,809.56	5.39	31.05	81.69	−0.228
*EgGUX22*	Eucgr.L00250.1.v2.0	338	38,883.79	5.54	30.36	84.53	−0.249
*EgGUX23*	Eucgr.L01804.1.v2.0	320	37,295.73	5.70	39.17	72.16	−0.431

## Data Availability

All data generated or analyzed during this study are included in this article and its Appendix A. The gene sequence, CDS sequence, and GFF annotation information of all *EgGUX* gene families are included in the Appendix A.

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
