# Peer review of "Genome-Wide Identification and Functional Analysis of the GUX Gene Family in Eucalyptus grandis"

_ijms, 2024, doi:10.3390/ijms25158199_

Round 1
Reviewer 1 Report
Comments and Suggestions for Authors
Interesting manuscript about GUX gene family in eucalyptus. Only a moderate revision is required before publication in IJMS:
Around the whole manuscript, the term in vitro and the name of genes must be in italic.
Plant material is poor. Description of eucalyptus clone must be clarified and referenced. In addition, the assayed experimental design should also be clarified with a clear description of the assayed samples and the used replications.
In silico and bioinformatic analysis should be clarified together with the new experimental results.
In general Discussion section should be completed with new biological information about the xylan synthesis and other physiochemical components in eucalyptus.
A new Conclusion section is required including main implications of the obtained results from a breeding or production point of view.
Comments on the Quality of English LanguageOnly minor revisions are required
Reviewer 2 Report
Comments and Suggestions for Authors
The ms. by Li and colleagues is a nice piece of work dealing with the reconstruction of the phylogenetic relationship of the GUX family genes in E. grandis. This is associated with a functional in vitro analysis of the encoded proteins and with an assessment of the transcription rate in response to abiotic stress and phytohormones. The article is well-written and to the point, making it easy and pleasant to read.
There is, however, the point of the Figures, which, at least in my copy are of low quality, thus my request is to provide higher resolution pictures for Figures from 1 to 7. This is also important because of the relevance that these figures have for the manuscript.
Also, at lines 255-256, the Authors state that “To analyze the genome-wide duplication of EgGUXs, members of the E. grandis GUX gene family were subjected to covariance analysis (Fig. 4a)”. However, no further mention of the analysis of covariance is to be found, please describe it in the text or in the legend and give a better description of the variables used in the M&M section, because the way it is written is the only obscure part of the paper.
After addressing these minor points, my recommendation is that this paper be published without any further revision.
Round 2
Reviewer 1 Report
Comments and Suggestions for Authors
Authors have revised correctly the manuscript
Author Response
Thank you for your valuable comments to make the manuscript more readable and scientific.